# Cannabinoid Receptors in the Horse Lateral Nucleus of the Amygdala: A Potential Target for Ameliorating Pain Perception, Stress and Anxiety in Horses

**DOI:** 10.3390/ijms26157613

**Published:** 2025-08-06

**Authors:** Cristiano Bombardi, Giulia Salamanca, Claudio Tagliavia, Annamaria Grandis, Rodrigo Zamith Cunha, Alessandro Gramenzi, Margherita De Silva, Augusta Zannoni, Roberto Chiocchetti

**Affiliations:** 1Department of Veterinary Medical Sciences (UNI EN ISO 9001:2008), University of Bologna, 40126 Bologna, Italy; cristiano.bombardi@unibo.it (C.B.); giulia.salamanca2@unibo.it (G.S.); annamaria.grandis@unibo.it (A.G.); rodrigozamithcunha@gmail.com (R.Z.C.); margiedesilva@gmail.com (M.D.S.); augusta.zannoni@unibo.it (A.Z.); 2Department of Translational Medicine and for Romagna, University of Ferrara, 40121 Ferrara, Italy; 3Department of Veterinary Medicine, University of Teramo, 64100 Teramo, Italy; ctagliavia@unite.it (C.T.); agramenzi@unite.it (A.G.)

**Keywords:** cannabinoid receptor 1, cannabinoid receptor 2, equine, peroxisome proliferator-activated receptor gamma, transient receptor potential vanilloid type 1

## Abstract

The amygdala is composed of several nuclei, including the lateral nucleus which is the main receiving area for the input from cortical and subcortical brain regions. It mediates fear, anxiety, stress, and pain across species. Evidence suggests that the endocannabinoid system may be a promising target for modulating these processes. Cannabinoid and cannabinoid-related receptors have been identified in the amygdala of rodents, carnivores, and humans, but not in horses. This study aimed to investigate the gene expression of cannabinoid receptors 1 (CB1R) and 2 (CB2R), transient receptor potential vanilloid 1 (TRPV1), and peroxisome proliferator-activated receptor gamma (PPARγ) within the lateral nucleus of six equine amygdalae collected post mortem from an abattoir using quantitative real-time PCR, cellular distribution, and immunofluorescence. mRNA expression of CB1R and CB2R, but not TRPV1 or PPARγ, was detected. The percentage of immunoreactivity (IR) was calculated using ImageJ software. Cannabinoid receptor 1 immunoreactivity was absent in the somata but was strongly detected in the surrounding neuropil and varicosities and CB2R-IR was observed in the varicosities; TRPV1-IR showed moderate expression in the cytoplasm of somata and processes, while PPARγ-IR was weak-to-moderate in the neuronal nuclei. These findings demonstrate endocannabinoid system components in the equine amygdala and may support future studies on *Cannabis* spp. molecules acting on these receptors.

## 1. Introduction

The amygdala, or amygdaloid complex, is an anatomically and functionally heterogeneous structure [1] located in the medial part of the temporal lobe [2]. Current neurochemical, functional, and developmental data suggest that the amygdala is composed of pallial and subpallial structures. The pallial amygdala is composed of deep and superficial nuclei. The deep pallial nuclei include the basolateral amygdala, the anterior amygdaloid area, and the amygdalohippocampal area. The basolateral amygdala (BLA) includes the lateral, basal, and accessory basal nuclei [2,3]. The lateral nucleus is the main receiving structure of the amygdala. It collects most of the input from other brain regions and relays information to other amygdala nuclei and higher brain areas [4,5,6]. The lateral nucleus plays a crucial role in the formation of emotional memories. It is also involved in generating appropriate behavioral responses to salient sensory and emotionally arousing stimuli, including nociceptive input [7,8,9,10,11,12,13]. The increasing interest in amygdala function and dysfunction, especially in neuropsychiatric disorders, has stimulated many studies. These investigations have aimed to clarify how emotional stimuli are processed within the amygdala, both in terms of connectivity and at the cellular and molecular levels. In horses, the amygdala is relatively well developed [14] and may be of great importance in this prey species, which is particularly alert to signs of threat and responds with rapid avoidance behavior [15]. However, little is known about the neurochemical properties of the equine amygdala [14,15].

The endocannabinoid system (ECS) consists of cannabinoid receptors (CB1R, CB2R) and cannabinoid-related receptors, their ligands (anandamide, AEA; 2-arachidonoylglycerol, 2-AG; and other ligands), and the enzymes necessary for their synthesis and degradation. Cannabinoid-related receptors include transient receptor potential (TRP) channels, G protein-coupled receptors (GPRs), nuclear peroxisome proliferator-activated receptors (PPARs), and serotonin receptors [16,17,18,19,20]. In the past decade, the ECS has emerged as a key neuromodulatory system regulating synaptic activity within and between the amygdaloid nuclei. Through its interaction with other systems, such as the opioid system, the ECS influences vital physiological functions and emotional states. The ECS is probably the most powerful and ancient modulatory system capable of influencing every conscious thought and feeling [16,17,18,19,20]. Molecular components of the ECS are abundantly expressed in the mammalian amygdaloid complex. This is consistent with its important role in modulating emotional behavior, learning, and stress responses [21,22].

Improving our morphofunctional knowledge of the equine amygdala may enhance the understanding of this species’ behavioral domain. This includes cognition, learning, stereotypies, separation anxiety, temperament-related welfare, and psychopharmacology. To date, no studies have been published regarding the expression of cannabinoid and cannabinoid-related receptors in the equine amygdaloid complex. Therefore, the present study was designed to evaluate the mRNA of *Cnr1*, *Cnrr2*, *PPARγ*, and *TRPV1* gene expression on fixed amygdala samples, and to immunohistochemically localize these receptors in the equine amygdala, particularly in the extensively studied lateral nucleus of the amygdala.

## 2. Results

### 2.1. Quantitative Real-Time PCR (RT-PCR) for Cnr1, Cnr2, TRPV1, and PPARγ

The reference utilized for data normalization was glyceraldehyde-3-phosphate dehydrogenase (GAPDH), because hypoxanthine phosphoribosyltransferase 1 (HPRT) was undetectable and actin was characterized by a huge variability in the samples. Quantitative real time PCR showed that the *Cnr1* and *Cnr2* transcripts were detected in all the 4% paraformaldehyde-fixed amygdala samples (*n* = 6) (Figure 1). On the contrary, *TRPV1* and *PPARγ* were undetectable.

### 2.2. Immunofluorescence

All the receptors expressed immunoreactivity in the lateral nucleus of the horse amygdaloid complex with, however, different degrees of intensity.

#### 2.2.1. CB1R

Cannabinoid receptor 1 immunoreactivity (CB1R-IR) was brightly expressed by the neuropil which extended within the lateral nucleus. The CB1-IR neuropil was characterized by the classic basket shape surrounding the bodies of other neurons (Figure 2). The calculated percentage of immunoreactivity was 4.96 ± 3.16%.

#### 2.2.2. CB2R

Cannabinoid receptor 2 immunoreactivity (CB2R-IR) was moderately expressed by the neuropil within the lateral nucleus. The neuropil expressing CB2-IR was characterized by a diffuse punctate appearance (Figure 3). The calculated percentage of CB2R-IR was the lowest (as compared to the other receptors), 3.22 ± 1.84%.

#### 2.2.3. TRPV1

Transient receptor potential vanilloid 1 immunoreactivity (TRPV1-IR) was moderately expressed by the neuronal bodies and processes within the lateral nucleus; TRPV1-IR was distributed in the neuronal cell bodies and processes, which formed a reticular pattern (Figure 4). The calculated percentage of TRPV1-IR was the highest (as compared to the other receptors) 10.72 ± 5.92%.

#### 2.2.4. PPARγ

Peroxisome proliferator-activated receptor gamma immunoreactivity (PPARγ -IR) was brightly expressed in the nuclei of neuronal cells and glial cells (Figure 5). Virtually all the NeuroTrace^®^-labeled neuronal cell bodies showed PPARγ -IR; the calculated percentage of PPARγ -IR was 9.03 ± 3.83%

Figure 6 shows the percentages of CB1R, CB2R, TPRV1, and PPARγ immunostaining within the lateral nucleus of the horse amygdaloid complex.

## 3. Discussion

The mammalian amygdala is composed of more than a dozen nuclei which have distinctive connections [4,23]. The BLA consists of three main nuclei, the lateral nucleus, the basal nucleus, and the accessory basal nucleus, each of which has several subdivisions [4]. The lateral nucleus has a cortical-like organization and contains two main neuronal types. These include spiny glutamatergic pyramidal projection neurons and spine-sparse or spine-free GABAergic nonpyramidal neurons [24,25,26]. The lateral nucleus is a key integrative site for emotional processing and pain modulation, making it a strategic target for investigating the distribution of cannabinoid and cannabinoid-related receptors in the horse.

In general, cannabinoid receptors are located densely in the animal and human brain areas [27] as in other central and peripheral nervous system areas of the horse, such as trigeminal ganglion, dorsal root ganglia, and also in the synovial membrane [28,29,30,31].

Relatively few studies have explored the distribution and expression of cannabinoid receptors in the amygdaloid complex across species to date. This is the first study to document the presence, both at molecular and cellular level of CB1, CB2, TRPV1, and PPARγ receptors in the lateral nucleus of the amygdaloid complex in horses. Most of the available literature has focused on CB1 receptors, particularly in rodent models. In rats and mice, CB1R has been shown to be strongly expressed in the basolateral amygdala (BLA), particularly in the lateral and basal nuclei. Katona et al. [32] reported that CB1R is predominantly localized presynaptically on the terminals of a specific subset of GABAergic interneurons that co-express cholecystokinin (CCK). Using electron microscopy, Fitzgerald et al. [33] confirmed that CB1R is concentrated in large, morphologically complex presynaptic boutons in the BLA and quantified a considerable number of CB1-positive axon terminals. Similarly, McDonald and Mascagni [21] identified CB1R immunoreactivity in weakly stained pyramidal neurons and intensely labelled multipolar aspiny interneurons; the latter correspond to CCK-positive GABAergic cells. Although less extensive, some evidence is also available in humans. Glass et al. [34] used quantitative autoradiography with the radioligand [^3^H]CP55,940 to demonstrate high binding site density for cannabinoid receptors in the human amygdaloid complex. Furthermore, Wang et al. [35] reported strong CB1R mRNA expression in the basal nuclear group of the amygdala and more moderate levels in the cortical and medial subdivisions using in situ hybridization in fetal human brains. Limited morphological and molecular data on cannabinoid receptors are also available in the marmoset amygdaloid complex [36]. In the dog, Freundt-Revilla et al. [37] investigated the central nervous system (CNS) but did not specifically examine the amygdala. Together, these findings support the idea that the distribution of CB1Rs in the amygdala is a conserved feature across mammalian species, and they emphasize the importance of our data in a broader comparative neuroanatomical context.

In contrast to CB1R, far fewer studies have investigated the distribution of CB2, TRPV1, and PPARγ receptors in the amygdaloid complex, and most of the available data come from rodent models. Regarding CB2R, immunohistochemical evidence in the rat brain revealed widespread expression of this receptor, including in the amygdala, where CB2 immunoreactivity was detected in putative neuronal cell bodies, glial processes, and neuronal fibers [38]. As for TRPV1, autoradiographic studies using [^3^H]Resiniferatoxin demonstrated its presence throughout the CNS of the rat, including the amygdala [39], and a subsequent immunohistochemical investigation confirmed the expression of VR1 (TRPV1) in neurons of the central amygdaloid nucleus [40]. PPARγ expression has also been documented in the mouse amygdala, where immunohistochemical analyses revealed that this receptor is present in approximately 60% of GABAergic and 40% of glutamatergic neurons [41]. A previous study further confirmed the high expression of PPARγ in the basolateral amygdala and its substantial colocalization with GABAergic neurons, suggesting a potential role in modulating inhibitory neurotransmission [42].

Regarding the gene expression of *Cnr1*, *Cnr2*, *TRPV1*, and *PPARγ* receptors in the amygdaloid complex on animal model, the data are relatively scarce. The transcripts of *Cnr1* and *Cnr2* were detected in rats [43,44] and in the bovine [45] amygdaloid complex while *TRPV1* and *PPARγ* were detected only in rodents [41,43].

Typically, enhanced CB1R activation reduces anxiety, while its inhibition produces anxiogenic effects [46,47]. For this reason, compounds that modulate cannabinoid receptor activity may help to reduce anxiety and represent promising targets for future anxiolytic therapies in equine medicine. Since cannabidiol (CBD) does not act directly on CB1R [48], but rather as a hydrolyzer of fatty acid amide hydrolase (FAAH) which is the membrane enzyme hydrolyzing AEA [49], it could potentially reduce anxiety [27].

Fear extinction is a type of inhibitory learning. In this process, new associations compete with previously established fearful ones, suppressing fear responses and modulating fear-related neural circuits [50]. In rodents, fear extinction activates the ECS through projections from the prefrontal cortex to the BLA [51]. Glutamate released from prefrontal neurons stimulates cannabinoid production in the BLA, which then acts presynaptically on CB1 receptors. This suppresses further glutamate release and promotes the extinction of the conditioned fear response [52]. During stress, the production of endocannabinoids may modulate emotional responses by altering amygdalar neurotransmission [53]. Typically, anxiety elevates endocannabinoid tone, and endocannabinoids in turn reduce anxiety, as demonstrated by Marsicano et al. [54]. In addition, endocannabinoids appear to be crucial for the extinction of aversive memories and this process is likely to be mediated by the amygdala [55]. In addition to its role in fear extinction, the ECS has also been implicated in fear-conditioned analgesia, a phenomenon whereby pain perception is suppressed in the presence of a fear-related context. In rodents, Rea et al. [56] showed that CB1R activation in the BLA is required for fear-induced analgesia. Administration of a CB1R antagonist directly into the BLA abolished the antinociceptive effect triggered by contextual fear. Rubino et al. [57] showed that treatment with the phytocannabinoid ∆9-THC, an agonist of CB1R, significantly decreased c-Fos levels in the amygdala, and that this effect was reversed by the CB1 inverse agonist AM251. Taken together, these findings underscore the central role of CB1R in modulating emotional and nociceptive processes through the amygdaloid circuitry. The identification of CB1R protein expression within the lateral nucleus of the equine amygdala in the present study supports the hypothesis that similar regulatory mechanisms of fear, anxiety, and pain may be conserved in horses, providing a neuroanatomical foundation for the potential therapeutic use of cannabinoid-based compounds in equine behavioral medicine.

Although traditionally considered a peripheral receptor mainly involved in immune function, CB2R expression has been increasingly reported in the CNS, including neurons and glial cells, challenging the early notion of its absence from the brain [58,59]. In the present study, CB2R-IR was observed in varicosities within the lateral nucleus of the amygdala, and Cnr2 gene expression was detected by RT-PCR, supporting the presence of this receptor at both protein and transcript levels. These findings are in line with previous studies showing CB2R localization in the neuropil and dendritic compartments of several brain regions, including the amygdala, hippocampus, and substantia nigra of rodents [60,61]. Ultrastructural studies have shown that CB2R can be found postsynaptically, often on dendritic membranes and the rough endoplasmic reticulum. This suggests endogenous synthesis and potential integration into synaptic signaling pathways [60]. Although CB2R-IR expression is lower than that of CB1R, CB2R in the CNS appears to play a modulatory role, particularly under pathological conditions, such as neuroinflammation, stress, and neuropsychiatric disorders [61,62]. Functionally, CB2R activation in the brain has been associated with the regulation of microglial activity, immune response modulation, and neuroprotection [59]. Moreover, behavioral studies in animal models have suggested that CB2R agonists, such as β-caryophyllene, exert anxiolytic and antidepressant effects, reinforcing the potential involvement of CB2R in the modulation of emotional states [63]. Several lines of evidence support a functional role for CB2R within the amygdala. Its expression has been documented under both physiological and stress-related conditions [64,65], and CB2R activation in the central nucleus has been shown to mediate cannabinoid-induced antinociceptive effects [66]. In the medial amygdala, early-life stimulation of CB2R influences neuronal morphology and contributes to the development of social behaviors [67]. Although most of the research has focused on CB1R within the BLA, recent findings have suggested that CB2R activation in this region may also modulate stress-induced alterations in synaptic plasticity and affective responses. Specifically, intra-BLA administration of the CB1R/CB2R agonist WIN55,212-2 reversed stress-impaired hippocampal–accumbens long-term potentiation, suggesting that CB2R could contribute to the amygdaloid regulation of stress-related neural circuits [68]. While direct evidence of CB2R expression in the BLA, and especially in the lateral nucleus, remains limited, the present findings aligned with the broader observations of CB2R functional relevance across the amygdaloid complex [58]. These data support the hypothesis that CB2R may also modulate emotional and sensory processing in these subregions.

The transient receptor potential vanilloid type 1 receptor is a non-selective cation channel known for its role in nociception and thermoregulation. However, increasing evidence has revealed its expression in the brain areas involved in emotional regulation, including the amygdala [69]. In particular, TRPV1 has been identified in the lateral nucleus of the amygdala, where it modulates synaptic plasticity and long-term potentiation, thereby influencing fear memory formation in mice [69]. Although the precise physiological role of TRPV1 in the amygdala remains unclear, functional studies in TRPV1 knockout mice demonstrated reduced anxiety-like behavior, and diminished fear conditioning and stress-sensitization, additionally implicating this receptor in emotional processing circuits [70,71]. Transient receptor potential vanilloid type 1 signaling intersects with the ECS since it can be activated by anandamide, a key endocannabinoid molecule [72]. This convergence has prompted the development of dual FAAH/TRPV1 inhibitors which have shown dose-dependent antidepressant-like effects in rodent models of depression [73], underscoring the therapeutic relevance of this pathway.

TRPV1 is not only involved in emotional regulation but also plays a key role in pain perception and neurogenic inflammation. When activated on peripheral sensory terminals, TRPV1 allows a rapid influx of calcium and sodium ions. This leads to the release of neuropeptides, including substance P, neurokinin A (NKA), and calcitonin gene-related peptide (CGRP) [74]. In addition to peripheral effects, TRPV1 activation in primary sensory neurons also modulates central nociceptive processing, supporting its role in the higher-order processing of nociceptive signals [74]. At the molecular level, TRPV1 protein has been shown to localize to the plasma membrane, cytoplasmic vesicles, and membranes of the endoplasmic reticulum in neurons, compartments which support its involvement in both calcium influx and intracellular calcium release [75] with some studies also describing its presynaptic localization in amygdaloid circuits [76]. These data are in line with the immunofluorescence results in the present study which revealed TRPV1 expression predominantly in the neuronal somata and neuropil within the lateral nucleus of the equine amygdala, with notable cytoplasmatic distribution patterns. The effect of TRPV1 activation on synaptic transmission varies depending on the brain region. According to Edwards et al. [77], TRPV1 can enhance or inhibit glutamatergic signaling, depending on the specific neural context. Altogether, these findings support a multifaceted role for TRPV1 as a molecular integrator of nociceptive and emotional signals and highlight its conserved expression and functional relevance within the amygdaloid circuitry across mammalian species, including the horse, as demonstrated by the present study.

Peroxisome proliferator-activated receptor gamma is a nuclear receptor traditionally associated with metabolic regulation and anti-inflammatory signaling; however, it has also been implicated in CNS functions, including emotion, stress, and pain modulation. It has been shown to be expressed in rat, mouse, monkey, and human neurons, astrocytes, and rat microglial and oligodendrocyte cell cultures where it contributes to mitochondrial protection, redox balance, and modulation of neuroinflammatory responses [78,79,80,81,82,83,84]. In the present study, PPARγ-IR was detected in the lateral nucleus of the equine amygdala, localized to the nuclear compartment of both neurons and glial fibrillary acidic protein (GFAP)-positive glial cells. This distribution is consistent with its role as a ligand-activated transcription factor. When activated by selective agonists like rosiglitazone or pioglitazone, PPARγ forms a heterodimer with the retinoid X receptor. This complex then binds to peroxisome proliferator response elements. It has been shown to reduce anxiety and depression-like behavior, attenuate fear expression, and alleviate mechanical allodynia and thermal hyperalgesia by means of mechanisms involving both central and peripheral pathways [85,86]. Taken together, these findings reinforce the view that PPARγ plays a modulatory role in emotional and nociceptive processing. The authors’ identification of PPARγ in both neurons and GFAP-positive glial cells of the lateral amygdaloid nucleus in the horse adds novel comparative neuroanatomical evidence to this growing body of the literature and supports a conserved role for PPARγ in the regulation of limbic functions across mammalian species.

Although the effects of cannabinoids on emotion and memory are well documented, little is known about how they influence neuronal networks in the equine brain, particularly within the amygdala. In the present experiment, the authors focused on the lateral nucleus which is the best-described region of the mammalian amygdala. The present investigation demonstrated the presence of the CB1, CB2, TRPV1, and PPARγ receptors in the equine amygdaloid complex for the first time. Furthermore, the present investigation showed that there was a very high density of CB1R axon terminals forming pericellular baskets presumably contacting the pyramidal somata, suggesting that cannabinoids may reduce tonic GABAergic inhibitory control over pyramidal cells in the lateral nucleus, as demonstrated by previous immunohistochemical and electrophysiological findings [32]. The distribution of other cannabinoid receptors in neuropilar and somatal structures supports strong cannabinoid modulations on amygdala activities. Consistent with the present findings, the BLA of primates [22] and rodents [21] contains many CCK-IR axon terminals containing the CB1R. These axons form a pericellular basket which contacts the somata of putative pyramidal cells and may be critical for fear expression and extinction [21,22]. Regarding the transcripts of the genes of interest, gene expression for *Cnr1* and *Cnr2*, consistent with the protein (CB1 and CB2) data was detected. In contrast, *TRPV1* and *PPARγ* transcripts were not detected in the fixed amygdala samples. However, since the authors were able to detect these transcripts in non-fixed samples (preliminary data) and in line with the protein expression data, they suggested that the absence of *TRPV1* and *PPARγ* transcripts in the fixed samples described in this manuscript was due to the known negative effect of formaldehyde fixation on RNA integrity, as has been well documented in the literature [87,88].

Despite the valuable insights provided, some limitations of the present study must be acknowledged. The sample size (*n* = 6), although reflective of the inherent difficulty in obtaining well-preserved equine brain specimens, limits the generalizability of the findings. Additionally, the wide age range of the animals included (3–25 years) could influence the anatomical and molecular features of the CNS. Although this variability mirrors the heterogeneity encountered in real-world conditions, future studies should include larger, more stratified cohorts, by both age and sex, to clarify whether these factors impact receptor expression or localization within the amygdala. Expanding the sample base would also help identify potential interindividual variability in the ECS. Comparative analyses across species could further elucidate the evolutionary conservation and functional roles of cannabinoid and cannabinoid-related receptors in limbic circuits. Moreover, the adoption of advanced techniques such as in situ hybridization, electron microscopy, or single-cell transcriptomics may refine our understanding of their cellular and subcellular distribution. In particular, it would be of great interest to investigate receptor expression in horses exhibiting stress- or anxiety-related behavioral disorders and in individuals experiencing pain, to explore their potential involvement in pathological states and their value as therapeutic targets.

## 4. Materials and Methods

The brains of six healthy horses, between 3 and 25 years of age, 4 female and 2 male, 4 slaughtered for human consumption, and 2 died from causes unrelated to the central nervous system, were removed from the skull post mortem. The breeds included 1 Avelignese, 1 Italian thoroughbred and 2 half-breeds, and 2 warmbloods. As the abattoir procedure involves the longitudinal mechanical opening of the skull along its dorsal part, twelve halves of the brain were quickly removed from the skull. This procedure usually does not damage the most ventral part of the brain, the piriform lobe in which the amygdala is located. After severing the connections to some of the cranial nerves, especially the optic nerve which anchors the diencephalon to the sphenoid bone, the brains were removed from those skulls in which the entire neural anatomy of the brain was better preserved. The brains used in the current study came from a number of the same horses used in a previous study [28]. Horses showing no neurological symptoms or behavioral problems were considered to be healthy on the basis of a summary clinical examination. In accordance with Directive 2010/63/EU of the European Parliament and of the Council of 22 September 2010 regarding the protection of animals used for scientific purposes, Italian legislation (D. Lgs. n. 26/2014) does not require approval by the competent authorities or ethics committees as this study did not influence any therapeutic decisions.

### 4.1. RNA Isolation and Quantitative Real Time PCR (RT-PCR) for Cnr1, Cnr2, PPARγ and TRPV1

For gene expression analysis, total mRNA extraction was carried out using a RNeasy FFPE Kit (Qiagen, Hilden, Germany) with a few modifications. Fifty mg of formalin fixed amygdala samples (*n* = 6) were washed in phosphate-buffered saline (PBS) and cut finely with disposable scalpel. After adding 240 µL of Buffer PKD (Qiagen, Hilden, Germany), the samples were briefly centrifuged, and Proteinase k (10 µL) was added. The samples were incubated at 56 °C for 4 h; every 30 min, the samples were mixed/homogenized using a micro-pestle (Eppendorf, Hamburg, Germany). The samples were then incubated at 80 °C for 15 min; the manufacturer’s protocol was followed up to elution (20 µL). After spectrophotometric quantification, the total RNA (500 ng) was reverse transcribed to cDNA using 5× iScript RT Supermix (Bio-Rad Laboratories Inc., Hercules, CA, USA) at a final volume of 20 μL. To evaluate the gene expression profiles, RT-qPCR was carried out in a CFX96 thermal cycler (Bio-Rad Laboratories Inc.) using SYBR green detection to target the genes. Specific primers for horses [28] were used to evaluate the gene expression for the interest gene (IG) Cannabinoid receptors 1 and 2 (*Cn1r* and *Cn2r*) and potential vanilloid type 1 (*TRPV1*). Primers for peroxisome proliferator-activated receptor gamma (PPARγ) were designed on a horse sequence (Accession Number KF788296) by using Beacon Designer 2.07 (Premier Biosoft International, Palo Alto, CA, USA) (For: 5′-CTAAAGAGCCTGAGAAAG-3′; Rev:5′-CCACTGAGAATAATGACA-3′): the specificity of the amplified *PPARγ* PCR products was confirmed by agarose gel electrophoresis and melting curve analysis. Regarding the reference genes (RG), *GAPDH*, *HPRT*, and *β-Act* (beta Actin) were based on horse sequences as previously reported [89]. All the amplification reactions were carried out in 20 µL and analyzed in duplicate; the reaction contained 10 µL of iTaq Universal SYBR Green Supermix (Bio-Rad Laboratories Inc., Boston MA, USA), 0.8 µL of the forward and reverse primers (5 mM each) of each target gene, 2 µL cDNA, and 7.2 µL of water. The real-time procedure included an initial denaturation period of 3 min at 95 °C, 40 cycles at 95 °C for 15 s, and 60 °C for 30 s, followed by a melting step with ramping from 55 °C to 95 °C at a rate of 0.5 °C/10 s. The specificity of the amplified PCR products was confirmed by agarose gel electrophoresis and melting curve analysis. The relative expression of the IGs were normalized based on the RGs. The relative mRNA expression of the genes tested was evaluated using the ΔCt method with ΔCt = (Ct _RG_ − Ct _IG_) which directly correlated with the expression level (ΔCt values very negative, lower expression; ΔCt values less negative higher expression).

### 4.2. Immunofluorescence

The brain halves were rapidly removed from the skulls of slaughtered horses and fixed for 48 h at 4 °C in 4% paraformaldehyde in phosphate buffer (0.1 M, pH of 7.2); the piriform lobes containing the amygdales were then dissected to obtain smaller pieces of tissue fixed for an additional 24 h. After a total fixation time of 72 h, the tissues were rinsed in PBS (0.15 M NaCl in 0.01 M sodium phosphate buffer, pH of 7.2) and stored at 4 °C in PBS containing 30% sucrose and sodium azide (0.1%). On the following days, the tissues were transferred to a mixture of PBS-30% sucrose-azide and Optimal Cutting Temperature (OCT) compound (Sakura Finetek Europe, Alphen aan den Rijn, The Netherlands) in a 1:1 ratio for an additional 24 h before being embedded in 100% OCT in Cryomold^®^. Prior to embedding, each sample (approximately 2 cm × 3 cm) was divided into three parts along its longitudinal axis (rostral, central, and caudal amygdala). In the current study, the central or rostral part of the amygdala (containing the lateral nucleus) was analyzed (Figure 7).

The sections were prepared by freezing the tissue in isopentane cooled in liquid nitrogen. Cryosections (14 μm thick) of the amygdala were cut on a cryostat and mounted on polylysinized slides (Thermo Fisher Scientific Inc., Waltham, MA, USA). The cryosections were hydrated in PBS and processed for immunostaining. To block non-specific binding, the sections were incubated for 1 h at RT (22–25 °C) in a solution containing 20% normal donkey serum (Colorado Serum Co., Denver, CO, USA), 0.5% Triton X-100g, and bovine serum albumin (1%) in PBS. The cryosections were incubated overnight at RT in a humidity chamber with anti-CB1R, -CB2R, -TRPV1, and -PPARγ; to identify the astrocytes, anti-glial fibrillary acidic protein (GFAP) antibodies were used (Table 1), diluted in a solution containing 20% normal donkey serum (Colorado Serum Co., Denver, CO, USA) 0.5% Triton X-100g (Sigma Aldrich, Milan, Italy), and bovine serum albumin (1%) (Sigma Aldrich, Milan, Italy) in PBS. After washing in PBS (3 × 10 min), the sections were incubated for 1 h at RT in a humidity chamber with secondary antibodies (Table 2) diluted in a solution containing 20% normal donkey serum (Colorado Serum Co., Denver, CO, USA), 0.5% Triton X-100g (Sigma Aldrich, Milan, Italy), and bovine serum albumin (1%) (Sigma Aldrich, Milan, Italy) in PBS. The cryosections were then washed in PBS (3 × 10 min) and processed for NeuroTrace^®^ labeling (1:200) (Thermo Fisher Scientific Inc., Waltham, MA, USA) as a pan-neuronal marker in PBS for 40 min, followed by mounting in buffered glycerol at pH of 8.6 with 4′,6-diamidino-2-phenylindole-DAPI (Santa Cruz Biotechnology, Santa Cruz, CA, USA).

### 4.3. Specificity of the Antibodies

The anti-CB1R, -TRPV1, and -PPARγ antibodies had already been tested by the present research group using Western blot (Wb) analysis on horse tissue [31,90]. The rabbit anti-CB2R antibody (PA1-744) had already been tested by Wb analysis on horse tissues [91]. For negative controls, the sections were prepared by omitting primary antibodies and then incubating them with the secondary antibodies. No stained cells were detected after omission of the primary antibodies.

### 4.4. Fluorescence Microscopy

The specimens were examined by the same observer using a Nikon Eclipse Ni microscope (Nikon Instruments Europe BV, Amsterdam, The Netherlands) equipped with the appropriate filter cubes. Images were captured using a DS-Qi1Nc digital camera and NIS Elements software BR 4.20.01 (Mountain View, Ottawa, ON, Canada) under identical exposure settings for all samples. Minor contrast and brightness adjustments were made using Corel Photo Paint 2017, while figure panels were prepared using Corel Draw 2017.

### 4.5. Semiquantitative and Quantitative Analysis of the Immunofluorescence

The immunoreactivity of the antibodies was evaluated, and their cellular localization within the amygdala was reported. The intensity of expression was evaluated semi-quantitatively as weak, moderate, and bright in images acquired at the same exposure times. The percentage of IR was evaluated quantitatively using ImageJ software (version 1.53e, NIH, USA) and expressed as the percentage mean ± standard deviation calculated on four sections (40× magnification) for each horse randomly selected within the rostro-caudal aspects of the lateral nucleus, identified based on anatomical landmarks as shown in Figure 7. For each field, a grayscale thresholding was applied manually using the “Threshold > Default > B&W > Dark background” method to identify IR signal, and the “Analyze > Measure” tool was used to calculate the percentage of IR over total field area selected manually avoiding the inclusion of autofluorescence parts, helped by the direct comparison with the original field. Background subtraction was performed uniformly across all images. The mean percentage and standard deviation of the IR signal were calculated for each animal. All image analyses were conducted by the same observer blinded to antibody identity to ensure consistency.

## 5. Conclusions

The present study is the first to demonstrate the genetic and immunohistochemical cellular distribution of the canonical CB1R, CB2R, and two cannabinoid-related receptors (TRPV1 and PPARγ) in the equine amygdaloid complex. Due to their cellular localization, these receptors may be the target of many drugs (endocannabinoids and endocannabinoid-related molecules, non-psychoactive phytocannabinoids, synthetic cannabinoids, and various agonists and antagonists) which could potentially be used to improve anxiety, stress, and pain in horses with behavioral problems and pathological conditions. Accumulating clinical evidence in horses supports the therapeutic potential of cannabinoid-based treatments for a variety of conditions, including chronic pain, inflammation, neuropathic sensitivity, and behavioral disturbances, highlighting the need for detailed anatomical studies such as the present one [92,93,94,95,96]. These results should hopefully encourage the development of new molecular and clinical studies to support the use of molecules already tested and used in humans and animals which could potentially reduce behavioral and pain-related problems in horses. Although the present study is mainly descriptive, it provides the first essential neuroanatomical framework for understanding the ECS in the equine amygdala, a brain region and species largely underrepresented in the literature. The use of post mortem tissue from abattoir-sourced samples imposed ethical and logistical constraints, which limited the implementation of functional or behavioral assays. Nonetheless, the combination of immunofluorescence and gene expression analysis enabled us to obtain reliable spatial and cellular data on receptor distribution. Although behavioral data could not be included due to the nature of the tissue source, the identification of ECS components at both gene and protein levels lays the groundwork for future studies integrating neuroanatomical findings with behavioral and physiological assessments in living animals. One limitation of this study is the lack of stratification by age or sex, which may affect receptor expression patterns. Due to the limited availability of equine brain samples, our primary goal was to establish a baseline distribution of ECS components in the amygdala. Future studies involving larger and more homogeneous cohorts will be needed to investigate potential age- or sex-related variations. These findings provide a foundation for future investigations into the functional role of ECS components in horses, especially in the context of emotional dysregulation, persistent stress, and pain-related conditions.

## Figures and Tables

**Figure 1 ijms-26-07613-f001:**
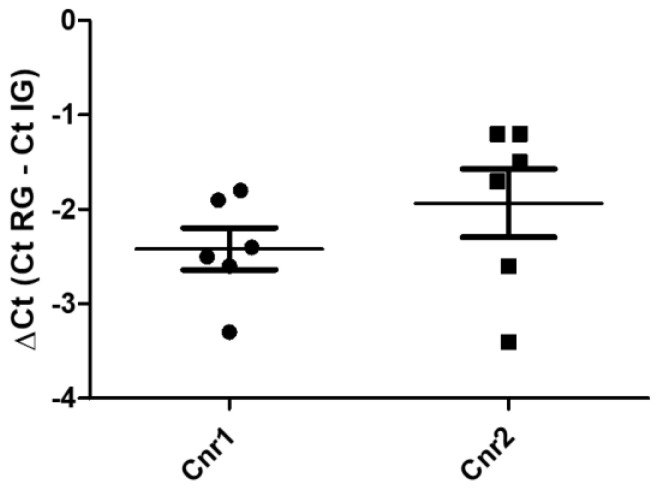
Gene expression of *Cnr1* and *Cnr2*, in the equine fixed amygdala samples. The results are presented as ΔCt = (Ct _RG_ − Ct _IG_), (ΔCt values less negative, higher expression). Symbols indicate individual animals. For each gene, mean ± standard deviation are indicated by horizontal bars. No significant statistical difference (*p* < 0.05) was observed between *Cnr1* and *Cnr2* gene expression (Student’s t-test, parametric test, *p* value = 0.131). RG = reference gene; IG = interest gene. Normal distributions were evaluated by means of Shapiro–Wilk tests (*Cnr1 p* = 0.5951; *Cnr2 p* = 0.1612), while the equality of variances in the two groups was assessed by means of Levene’s test (*F* value = 0.3058). GraphPad Prism v.8 (GraphPad Software Inc., San Diego, CA, USA).

**Figure 2 ijms-26-07613-f002:**
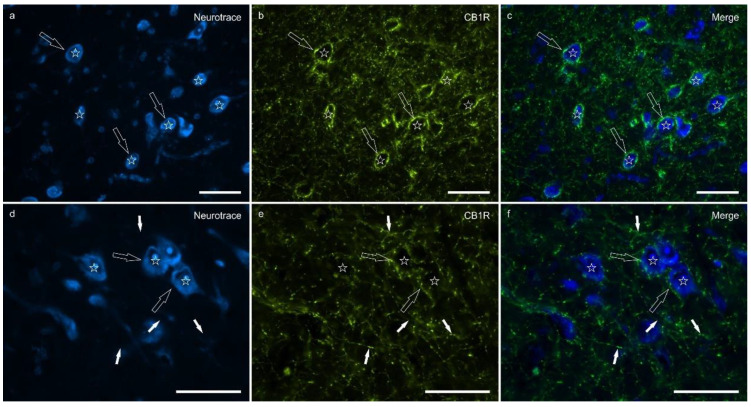
Photomicrographs of the cryosections of the lateral nucleus of the amygdala of a horse showing bright cannabinoid receptor type 1 (CB1R) immunoreactivity in thin nerve fibers (white fibers) of the neuropil and varicosities (open arrows). The stars indicate some neuronal cell bodies labeled with NeuroTrace (**a**,**d**) which were CB1R negative (**b**,**e**); the open arrows indicate baskets of bright CB1R immunoreactive varicosities encircling the neuronal cell bodies. The small white arrows (**d**–**f**) indicate neuropil fibers. (**c**,**f**) Merged image. Scale bar = 50 µm.

**Figure 3 ijms-26-07613-f003:**
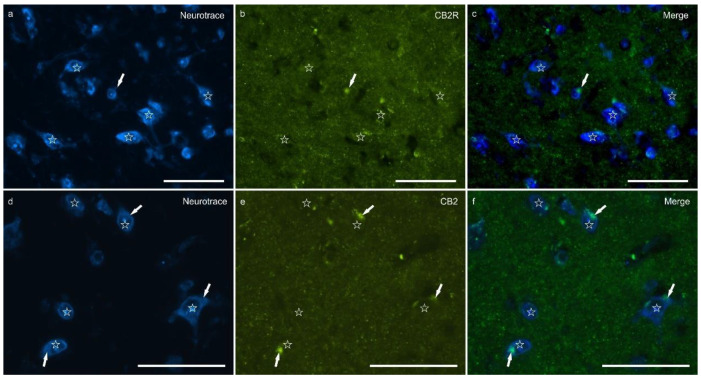
Photomicrographs of the cryosections of the lateral nucleus of the amygdala of a horse showing weak cannabinoid receptor type 2 (CB2R) immunoreactivity in the neuropil varicosities. The stars indicate some neuronal cell bodies labeled with NeuroTrace (**a**,**d**) which were CB2R negative (**b**,**e**); the white arrows indicate a deposit of autofluorescent pigment within the neuronal cell bodies. Note the diffuse punctate immunoreactivity. (**c**,**f**) Merged image. Scale bar = 50 µm.

**Figure 4 ijms-26-07613-f004:**
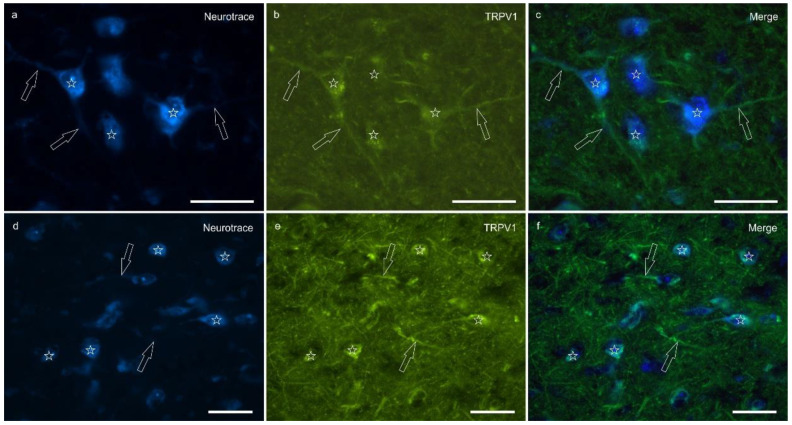
Photomicrographs of the cryosections of the lateral nucleus of the amygdala of a horse showing moderate-to-bright transient receptor potential vanilloid 1 (TRPV1) immunoreactivity (**b**,**e**) in NeuroTrace-labeled neuronal cell bodies (**a**,**d**) (stars), their nerve processes (open arrows) and the varicosities of the neuropil. (**c**,**f**) Merged image. Scale bar = 50 µm.

**Figure 5 ijms-26-07613-f005:**
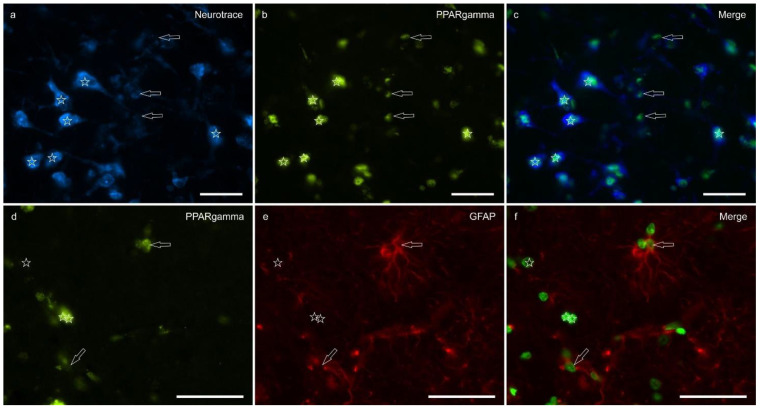
Photomicrographs of the cryosections of the lateral nucleus of the amygdala of a horse. (**a**–**c**) The stars indicate some NeuroTrace-labeled neurons (**a**) expressing bright nuclear peroxisome proliferator-activated receptor gamma immunoreactivity (PPARγ) (**b**). The open arrows indicate moderate PPARγ immunoreactivity in the smaller nuclei of the glial cells. (**d**–**f**) The open arrows indicate two glial fibrillary acidic protein (GFAP) immunoreactive glial cells co-expressing moderate nuclear PPARγ immunoreactivity (**d**). The arrows indicate the nuclei of GFAP negative cells. (**c**,**f**) Merged image. Scale bar = 50 µm.

**Figure 6 ijms-26-07613-f006:**
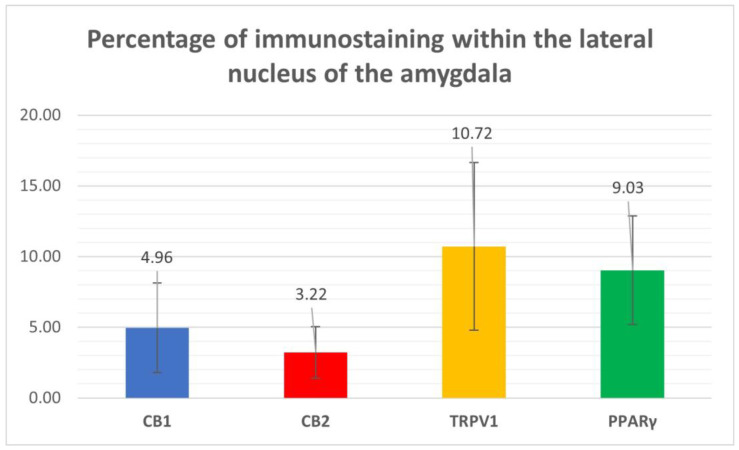
Percentages of immunostaining within the lateral nucleus of the horse amygdala of CB1R, CB2R, TPRV1, and PPARγ, calculated using ImageJ software.

**Figure 7 ijms-26-07613-f007:**
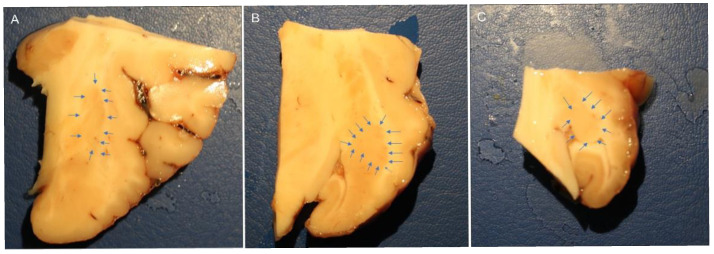
Macroscopic view of the right horse amygdaloid complex in a coronal section at the rostral (**A**), middle (**B**), and caudal (**C**) levels. The lateral nucleus boundary is indicated by the arrows.

**Table 1 ijms-26-07613-t001:** Primary antibodies used in the study.

Primary Antibody	Host	Code	Dilution	Source
CB1R	Rabbit	ab23703	1:200	Abcam
CB2R	Rabbit	PA1-744	1:500	Thermo Fisher
TRPV1	Rabbit	ACC-030	1:200	Alomone
PPARγ	Rabbit	Ab45036	1:300	Abcam
GFAP	Mouse	MAB360	1:1000	Millipore

Primary antibody Suppliers: Abcam, Cambridge, UK; Alomone, Jerusalem, Israel; Millipore Sigma, Burlington, MA, USA; Thermo Fisher Scientific, Waltham, MA, USA.

**Table 2 ijms-26-07613-t002:** Secondary antibodies used in the study.

Secondary Antibody	Host	Code	Dilution	Source
Anti-Rabbit 488	Donkey	A-21206	1:1000	Thermo Fisher
Anti-Mouse 594	Donkey	A-21203	1:500	Thermo Fisher

Secondary antibody supplier: Thermo Fisher Scientific, Waltham, MA USA.

## Data Availability

Data available on request from the corresponding author. The data are not publicly available due to file size limitations; the minimal dataset underlying the main figures and tables will be shared upon reasonable request.

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
