# Peer review of "Cannabinoid Receptors in the Horse Lateral Nucleus of the Amygdala: A Potential Target for Ameliorating Pain Perception, Stress and Anxiety in Horses"

_ijms, 2025, doi:10.3390/ijms26157613_

Round 1

Reviewer 1 Report

Comments and Suggestions for Authors

The amygdala is a small, almond-shaped structure in the brain, part of the limbic system, primarily involved in processing emotions, particularly fear and aggression, as well as memory and motivation. Cannabinoid receptors are a class of cell membrane receptors that are part of the larger endocannabinoid system in vertebrates. The two most well-characterized cannabinoid receptors are CB1 and CB2. CB1 receptors are predominantly found in the nervous system, while CB2 receptors are primarily located in the immune system as initially regarded. However, several studies indicated that this receptor is also expressed in the CNS. Impairment of fear extinction induced by OXA was associated with increased expression of CB2R in microglial cells of basolateral amygdala. Yale researchers found higher levels of cannabinoid receptors in the amygdala were associated with reduced reactions to pain and greater emotional numbing. In this study, author demonstrated the genetic and immunohistochemical cellular distribution of the canonical CB1R, CB2R, TRPV1, and PPARγ in the equine amygdaloid complex. Their findings indicated ECS components in the equine amygdala and may support future studies on Cannabis spp. molecules acting on these receptors. However, the experimental design is too simple, and the workload is too small.

1. In result section (Fig.1), I would recommend changing bar-graphs to reflect individual datapoints for transparency.

2. The results presented in the manuscript are all experimental results of molecular biology, lacking verification of animal behavior.

Author Response

The amygdala is a small, almond-shaped structure in the brain, part of the limbic system, primarily involved in processing emotions, particularly fear and aggression, as well as memory and motivation. Cannabinoid receptors are a class of cell membrane receptors that are part of the larger endocannabinoid system in vertebrates. The two most well-characterized cannabinoid receptors are CB1 and CB2. CB1 receptors are predominantly found in the nervous system, while CB2 receptors are primarily located in the immune system as initially regarded. However, several studies indicated that this receptor is also expressed in the CNS. Impairment of fear extinction induced by OXA was associated with increased expression of CB2R in microglial cells of basolateral amygdala. Yale researchers found higher levels of cannabinoid receptors in the amygdala were associated with reduced reactions to pain and greater emotional numbing. In this study, author demonstrated the genetic and immunohistochemical cellular distribution of the canonical CB1R, CB2R, TRPV1, and PPARγ in the equine amygdaloid complex. Their findings indicated ECS components in the equine amygdala and may support future studies on Cannabis spp. molecules acting on these receptors. However, the experimental design is too simple, and the workload is too small.

Response: We thank the Reviewer for this comment and for the opportunity to clarify the scope and rationale of our study. We acknowledge that the experimental design is focused on descriptive and qualitative analysis. However, this study represents the first investigation of the endocannabinoid system components in the equine amygdala, a species and brain region that are currently underrepresented in the literature. Working with equine tissue, especially brain samples obtained post mortem from abattoirs, involves logistical and ethical constraints that limit the use of experimental manipulations or in vivo designs. Nevertheless, we applied both qualitative real-time PCR and immunofluorescence to assess the gene and protein expression of CB1R, CB2R, TRPV1, and PPARγ, offering a morpho-molecular map that may serve as a foundation for future research. Although we did not include functional assays or larger-scale quantifications, we believe that our results provide novel and valuable baseline data for the understanding of ECS localization in the equine limbic system. We have revised the Conclusion section (lines 494-500) to better highlight the exploratory nature of this work and its potential as a starting point for behavioural and pharmacological studies in horses.

In result section (Fig.1), I would recommend changing bar-graphs to reflect individual datapoints for transparency.

We thank the Reviewer for the suggestion. We changed bar-graphs and figure description accordingly.

The results presented in the manuscript are all experimental results of molecular biology, lacking verification of animal behavior.

Response: We thank the Reviewer for highlighting this important aspect. We fully agree that integrating behavioural data would enhance the translational value of our findings. However, the present study was designed as an initial neuroanatomical and molecular mapping of ECS components in the equine amygdala, using tissue collected post mortem from an abattoir. This source of material inherently precluded the possibility of conducting behavioural assessments. Nonetheless, we believe that establishing the presence and cellular distribution of these receptors is a necessary first step toward designing future studies that may investigate their role in modulating emotional and behavioural responses in horses under different physiological or pathological conditions. We have revised the Conclusion section (lines 500-503) to better highlight the exploratory nature of this work and its potential as a starting point for behavioural and pharmacological studies in horses.

Reviewer 2 Report

Comments and Suggestions for Authors

I reviewed the manuscript entitled Cannabinoid receptors in the horse lateral nucleus of the amygdala: a potential target for ameliorating pain perception, stress and anxiety in horses.

I agree to accept this manuscript after major revision. 

1) Abstract, The author used many abbreviations here. In general, it is necessary to use abbreviations only when they occur three or more times. Otherwise, too many abbreviations will confuse readers. Please revise the abstract and the entire text according to this principle. For example, quantitative Real-Time PCR only appears once andendocannabinoid system only appears twice, and there is no need to use abbreviations. Simply use the full name.

2)  Keywords: The first letter of the first word of other keywords does not need to be capitalized, except for the first keyword and the keyword.

3) 2.1. PPARgamma, PPARγ, Appearing simultaneously in the text, it is suggested to uniformly change it to the latter. fixed amigdala samples (n=6), n should be italicized.

4) Figure 1. mean ± SD should change to mean ± standard deviation. P < 0.05, when it comes to statistics, P should be italicized. Please check and modify similar issues.

5) Figure 2. Scheme 50. m. Please verify if it is correct? Number 50 should not appear.

6) Figure 3. Scale bar = 50μm, The font and size do not meet the requirements, and there should be spaces between numbers and units. Please check and modify the entire text.

7) 4.1. RNA isolation and quantitative Real Time PCR (RT- PCR) for Cnr1, Cnr2, PPARγ and TRPV1  The first letter of each actual word in the secondary title needs to be capitalized. Therefore, the first letters of isolation and quantitative need to be capitalized. Check and revise the entire text.

8) 240ul should change to 240 uL. 30s should change to 30 s. Specific primers for horses (Zamith Cunha et al. 2023d) were used to evaluate the gene expression…The Reference’s format is incorrect, it should be changed to [15].

9) 48 hours should change to 48 h. 10 minutes should change to 10 min. To use international units instead of words, please check and modify similar issues.

10) In-depth studies are needed in the equine species since it is known that cannabinoids can have different effects on anxiety depending on the species [83]. In general, the conclusion section does not cite any literature.

11) The paper mentions the important role of ECS in regulating emotions, learning, and stress responses, but why choose horses as the research object? What is the unique value of studying horses compared to other species?

12) The sample includes horses of different ages (3-25 years old) and genders. Could these differences affect the experimental results? Have you considered the impact of age or gender on receptor expression?

13) The paper mentions using Student's t-test to compare gene expression, but does not specify whether normality and homogeneity of variance were tested for the data. Have non parametric tests been considered?

14) The paper mentions that CB1R may regulate emotions by inhibiting GABA release, but lacks direct evidence to support this mechanism. Is there any electrophysiological or behavioral experimental data to support it?

15) Does the study only use RT-PCR and immunofluorescence to further validate the results, considering other techniques such as in situ hybridization or single-cell sequencing?

16) Is a sample size of 6 horses sufficient to support the conclusion? Has efficacy analysis been conducted to determine the appropriate sample size?

17) Has the specificity of antibodies been validated through other methods, such as knockout models? Is there any reference to relevant validation literature?

18) Can the author consider splitting some sentences that are long and complex (such as discussing the first paragraph) to improve readability?

19) I have read all the references and found some issues. Ref 2, missing article number. Please add 68. Ref 6, Tursiops Truncatus is a Latin name and should be italicized. Ref 32, missing article number. Please add 1045030. The author also needs to check and revise according to the requirements of the journal.

20) This study investigated the gene expression and cellular distribution of cannabinoid receptors (CB1R, CB2R), TRPV1, and PPARɣ in the lateral nucleus of the equine amygdala using RT-PCR and immunofluorescence. mRNA expression of Cnr1 and Cnr2 was detected, while TRPV1 and PPARɣ transcripts were undetectable. CB1R immunoreactivity (IR) was localized to neuropil and varicosities, CB2R-IR to varicosities, TRPV1-IR to somata and processes, and PPARɣ-IR to neuronal nuclei. These findings confirm ECS presence in the horse amygdala, supporting future research on cannabinoid-based therapies for stress and pain modulation.

21) The biggest problem with this study is many details need to be modified and improved. Too many issues can make people feel that the author's attitude is not rigorous.The author must take them seriously and make necessary revisions.

Author Response

I reviewed the manuscript entitled Cannabinoid receptors in the horse lateral nucleus of the amygdala: a potential target for ameliorating pain perception, stress and anxiety in horses.

I agree to accept this manuscript after major revision.

Response: we thank the Reviewer for his/her appreciation and his/her valuable contribute to our work.

1) Abstract, The author used many abbreviations here. In general, it is necessary to use abbreviations only when they occur three or more times. Otherwise, too many abbreviations will confuse readers. Please revise the abstract and the entire text according to this principle. For example, quantitative Real-Time PCR only appears once and endocannabinoid system only appears twice, and there is no need to use abbreviations. Simply use the full name.

Response: we thank the Reviewer for his/her suggestions, we removed RT-PCR abb. (line 21) and ECS abb. (line 27).

2) Keywords: The first letter of the first word of other keywords does not need to be capitalized, except for the first keyword and the keyword.

Response: we thank the Reviewer for his/her suggestions. The keywords have been changed accordingly.

3) PPARgamma, PPARγ, Appearing simultaneously in the text, it is suggested to uniformly change it to the latter. fixed amigdala samples (n=6), n should be italicized.

Response: we thank the Reviewer for his/her suggestions. We made the changes accordingly.

4) Figure 1. mean ± SD should change to mean ± standard deviation. P < 0.05, when it comes to statistics, P should be italicized. Please check and modify similar issues.

Response: we thank the Reviewer for his/her suggestions. We made the changes accordingly.

5) Figure 2. Scheme 50. m. Please verify if it is correct? Number 50 should not appear.

Response: It is a mistake. We made the changes accordingly.

6) Figure 3. Scale bar = 50μm, The font and size do not meet the requirements, and there should be spaces between numbers and units. Please check and modify the entire text.

Response: we thank the Reviewer for his/her suggestions. We made the changes accordingly.

7) 4.1. RNA isolation and quantitative Real Time PCR (RT- PCR) for Cnr1, Cnr2, PPARγ and TRPV1. The first letter of each actual word in the secondary title needs to be capitalized. Therefore, the first letters of isolation and quantitative need to be capitalized. Check and revise the entire text.

Response: we thank the reviewer for his/her suggestions. We made the changes accordingly.

8) 240ul should change to 240 uL. 30s should change to 30 s. Specific primers for horses (Zamith Cunha et al. 2023d) were used to evaluate the gene expression…The Reference’s format is incorrect; it should be changed to [15].

Response: we thank the reviewer for his/her suggestions. We made the changes accordingly.

9) 48 hours should change to 48 h. 10 minutes should change to 10 min. To use international units instead of words, please check and modify similar issues.

Response: we thank the reviewer for his/her suggestions. We made the changes accordingly.

10) In-depth studies are needed in the equine species since it is known that cannabinoids can have different effects on anxiety depending on the species [83]. In general, the conclusion section does not cite any literature.

Response: we agree with the reviewer and we removed the sentence accordingly.

11) The paper mentions the important role of ECS in regulating emotions, learning, and stress responses, but why choose horses as the research object? What is the unique value of studying horses compared to other species?

Response: We thank the reviewer for his/her question. We highlighted the reasons beyond the choice of horse as research object in lines 48-50 of the introduction: “In horses, the amygdala is relatively well developed [14] and may be of great importance in this prey species, which is particularly alert to signs of threat and responds with rapid avoidance behavior [15].”

12) The sample includes horses of different ages (3-25 years old) and genders. Could these differences affect the experimental results? Have you considered the impact of age or gender on receptor expression?

Response: We thank the reviewer for this insightful comment. We acknowledge that both age and gender may influence receptor expression profiles in the brain. However, due to the limited availability of equine brain samples and the exploratory nature of this study, no stratification by age or sex was performed. Our primary objective was to map the presence and distribution of selected ECS components in the equine amygdala, which had not been previously described. The variability introduced by the heterogeneity of the sample is a limitation of the study, and we have now acknowledged this point in the revised Conclusion section (lines 504-508). Future studies with larger, stratified cohorts will be necessary to address possible age- or sex-related differences in receptor expression.

13) The paper mentions using Student's t-test to compare gene expression, but does not specify whether normality and homogeneity of variance were tested for the data. Have non parametric tests been considered?

Response: We thank the Reviewer for this helpful request. Normal distributions were evaluated by means of Shapiro–Wilk tests (Cnr1 P=0.5951; Cnr2 P=0.1612), while the equality of variances in the two groups was assessed by means of Levene’s test (F value =0.3058). According to the results (normal distribution and equality of variance), we performed a parametric Student's t. We added the data in the Fig. 1 legend.

14) The paper mentions that CB1R may regulate emotions by inhibiting GABA release, but lacks direct evidence to support this mechanism. Is there any electrophysiological or behavioral experimental data to support it?

Response: Katona et al., 2001 [74] used immunohistochemical and electrophysiological methods to demonstrate that the CB1 receptor is expressed by a specific population of interneurons in the rat basolateral amygdala, thereby inhibiting GABA release. We thank the Reviewer for this comment, we added the references in the discussion section (lines 339-340)

15) Does the study only use RT-PCR and immunofluorescence to further validate the results, considering other techniques such as in situ hybridization or single-cell sequencing?

Response: We thank the Reviewer for this valuable comment. Indeed, techniques such as in situ hybridization or single-cell RNA sequencing offer greater spatial and cell-type resolution for transcriptomic analysis. However, the current study was conducted on post mortem tissue from adult horses collected at an abattoir, which poses substantial constraints in terms of RNA preservation and tissue integrity, limiting the applicability of such high-resolution molecular techniques. Our primary goal was to establish the presence and anatomical distribution of ECS-related genes and proteins in the equine amygdala using complementary and reliable methods, quantitative RT-PCR for gene expression and immunofluorescence for cellular localization. We fully agree that future studies could benefit from the application of ISH or scRNA-seq to refine the understanding of receptor expression at the cellular level.

16) Is a sample size of 6 horses sufficient to support the conclusion? Has efficacy analysis been conducted to determine the appropriate sample size?

Response: We thank the Reviewer for this pertinent observation. We acknowledge that the sample size is limited and no formal power analysis was conducted prior to the study. However, this study was designed as an exploratory anatomical and molecular investigation using post mortem brain tissue from horses collected at an abattoir. The availability of suitable, well-preserved equine amygdalae is limited, and this imposes practical and ethical constraints on sample collection. Our aim was to provide a first baseline description of ECS component distribution in the equine amygdala, rather than to conduct inferential statistical analyses. The data were evaluated qualitatively, and where semi-quantification was performed (e.g., percentage of immunoreactive area), results were interpreted cautiously. We believe that our findings serve as a valuable starting point for future studies with larger, statistically powered cohorts to confirm and expand on our observations.

17) Has the specificity of antibodies been validated through other methods, such as knockout models? Is there any reference to relevant validation literature?

Response: We thank the Reviewer for this important observation. We fully acknowledge the relevance of antibody validation, particularly when working with a non-model species such as the horse. Although the use of knockout animals is a gold standard for antibody validation, its application in equine research is currently not feasible due to ethical and practical constraints, as genetically modified horses are not available. However, we have taken several measures to ensure antibody specificity and reliability in our study, which we mentioned in the M&M section:

  • For the molecular part, specific primers for Cnr1, Cnr2, Trpv1already used in equine tissue by Zamith et al. (2023), while primers sequence to detect transcript of PPARγ were reported in material and method section (paragraph 4.1). The specificity of the amplified PPARγ PCR products was confirmed by agarose gel electrophoresis and melting curve analysis (this sentence has been added in the text in material and method, paragraph 4.1).
  • For the immunofluorescence analysis, all antibodies were selected based on prior use and validation in the literature, especially on equine tissues. Specifically:
  • The anti-CB1R, -TRPV1, and -PPARγ antibodies had already been tested by our research group using Western blot analysis on horse tissues, as published in Chiocchetti et al. (2021) [80] and Galiazzo et al. (2021) [81].
  • The rabbit anti-CB2R antibody (PA1-744) had also been tested by Western blot on equine tissue in a separate study (Kupczyk et al., 2022) [82].
  • Negative controls were systematically included by omitting the primary antibody and incubating sections only with the secondary antibodies. No immunoreactivity was observed in these controls, confirming the specificity of the staining.

18) Can the author consider splitting some sentences that are long and complex (such as discussing the first paragraph) to improve readability?

Response: We thank the Reviewer for this helpful suggestion. We have carefully revised the manuscript to improve readability, especially in the Introduction and Discussion section. Several long and complex sentences have been split or rephrased to enhance clarity and fluency. We believe these edits improve the overall quality and accessibility of the text.

19) I have read all the references and found some issues. Ref 2, missing article number. Please add 68. Ref 6, Tursiops Truncatus is a Latin name and should be italicized. Ref 32, missing article number. Please add 1045030. The author also needs to check and revise according to the requirements of the journal.

Response: We thank the Reviewer for this correction. We now checked and revised all the references according to the requirements of the journal.

20) This study investigated the gene expression and cellular distribution of cannabinoid receptors (CB1R, CB2R), TRPV1, and PPARɣ in the lateral nucleus of the equine amygdala using RT-PCR and immunofluorescence. mRNA expression of Cnr1 and Cnr2 was detected, while TRPV1 and PPARɣ transcripts were undetectable. CB1R immunoreactivity (IR) was localized to neuropil and varicosities, CB2R-IR to varicosities, TRPV1-IR to somata and processes, and PPARɣ-IR to neuronal nuclei. These findings confirm ECS presence in the horse amygdala, supporting future research on cannabinoid-based therapies for stress and pain modulation.

Response: We thank the Reviewer for the accurate summary and positive assessment of our study.

21) The biggest problem with this study is many details need to be modified and improved. Too many issues can make people feel that the author's attitude is not rigorous.The author must take them seriously and make necessary revisions.

Response: We sincerely thank the Reviewer for the detailed and constructive feedback. We have carefully considered all suggestions and fully revised the manuscript to address every point raised. We trust that the updated version now reflects a more rigorous and accurate presentation of our work.

Reviewer 3 Report

Comments and Suggestions for Authors

I thank the authors for submitting this manuscript.

This type of work should be acknowledged, both for its originality and for the challenges involved in obtaining and processing such samples.

The methodology is clearly described, the introduction is well structured, and the results are presented concisely.

However, the Discussion section requires substantial revision.

In my opinion, the discussion is overly long (for example, lines 150–161 do not add relevant information) and tends to adopt the tone of an Introduction. There is also a lack of comparison between the current findings and those from previous studies in the same species. For instance, is the intensity and localization of CB1R immunohistochemical staining, as well as mRNA expression, similar or different between horses and humans? This kind of comparative analysis and the interpretation of its biological relevance is essential. The reader needs to understand how these results are applicable or biologically significant.

There is no need to discuss aspects that were not investigated or for which no results were obtained. Instead, the focus should be on a direct biological interpretation of the data presented.

The Discussion should also include a paragraph addressing the limitations of the study. While the difficulty in obtaining such samples is acknowledged, the conclusions that can be drawn from a study involving only six horses are inherently limited. I would also suggest reflecting on whether the age difference between the animals (e.g., 3 vs. 25 years old) could have anatomical or molecular implications at the central nervous system level.

I would also recommend including a brief paragraph suggesting future research directions that could stem from this study. It would be valuable to highlight which questions remain open and what further investigations—whether involving larger sample sizes, different age groups, or comparisons across species—could help deepen our understanding of CB1R expression and its biological relevance in the equine central nervous system.

Conclusion:

Lines 432–436 are speculative. Such considerations may be introduced in the Discussion but should be avoided in the Conclusion section.

Author Response

I thank the authors for submitting this manuscript.

This type of work should be acknowledged, both for its originality and for the challenges involved in obtaining and processing such samples.

The methodology is clearly described, the introduction is well structured, and the results are presented concisely.

Response: We sincerely thank the Reviewer for the positive and encouraging comments, which we greatly appreciate.

However, the Discussion section requires substantial revision.

In my opinion, the discussion is overly long (for example, lines 150–161 do not add relevant information) and tends to adopt the tone of an Introduction. There is also a lack of comparison between the current findings and those from previous studies in the same species. For instance, is the intensity and localization of CB1R immunohistochemical staining, as well as mRNA expression, similar or different between horses and humans? This kind of comparative analysis and the interpretation of its biological relevance is essential. The reader needs to understand how these results are applicable or biologically significant.

There is no need to discuss aspects that were not investigated or for which no results were obtained. Instead, the focus should be on a direct biological interpretation of the data presented.

Response: We thank the Reviewer for this valuable comment. In response, we have revised the initial part of the Discussion to remove the introductory tone and improve the focus on the biological interpretation of our experimental findings. We also reviewed and streamlined the text to reduce or eliminate sections not directly supported by our data. However, we believe that discussing the potential functional roles of the investigated receptors, based on findings from other species, remains important to place our results in a broader comparative neuroanatomical context. Therefore, we retained a limited, focused discussion of their known roles in the amygdala, particularly in relation to emotional and nociceptive processing, while ensuring that such content is clearly distinguished from our own findings. Furthermore, to address the Reviewer's request for a more explicit comparison with previous studies in the same or related species, we have added a short paragraph highlighting where present in the literature relevant interspecies differences and similarities in receptor localization and expression, although the lack of the same techniques and morphological approaches rise difficulties in precise comparisons (lines 165-209)

The Discussion should also include a paragraph addressing the limitations of the study. While the difficulty in obtaining such samples is acknowledged, the conclusions that can be drawn from a study involving only six horses are inherently limited. I would also suggest reflecting on whether the age difference between the animals (e.g., 3 vs. 25 years old) could have anatomical or molecular implications at the central nervous system level.

Response: We thank the Reviewer for the suggestion. We added a paragraph addressing the limitations of the study in the Discussion section (lines 353-360)

I would also recommend including a brief paragraph suggesting future research directions that could stem from this study. It would be valuable to highlight which questions remain open and what further investigations—whether involving larger sample sizes, different age groups, or comparisons across species—could help deepen our understanding of CB1R expression and its biological relevance in the equine central nervous system.

Response: We thank the Reviewer for the suggestion. We added a paragraph addressing the future research directions in the Discussion section (lines 361-370).

Conclusion:

Lines 432–436 are speculative. Such considerations may be introduced in the Discussion but should be avoided in the Conclusion section.

Response: We thank the Reviewer for this correction. We deleted the sentences from the Conclusion section.

Reviewer 4 Report

Comments and Suggestions for Authors

Critical Issues and Recommendations

  1. Limited Sample Size (n=6)
    Limitation:
    The small number of samples significantly reduces the statistical power of the study and limits the generalizability of the findings.

Recommendation:
Future studies should increase the sample size and include animals of varying age, sex, and physiological status to enhance the representativeness and robustness of the data.

  1. Use of Formaldehyde-Fixed Tissue for RT-PCR Analysis
    Limitation:
    The use of paraformaldehyde-fixed samples likely compromised RNA integrity, resulting in the inability to detect TRPV1 and PPARγ transcripts.

Recommendation:
Conduct parallel analyses on fresh tissue or tissue preserved in RNA-stabilizing reagents (e.g., RNAlater®), or extract RNA from frozen samples to obtain more reliable gene expression data.

  1. Discrepancy Between Gene Expression and Immunohistochemistry Data
    Limitation:
    The absence of TRPV1 and PPARγ mRNA despite evident protein immunoreactivity raises concerns about the reliability of the methods or sample preservation.

Recommendation:
Validate protein presence using complementary methods such as Western blotting on fresh tissue, and include rigorous negative and positive controls to confirm antibody specificity and immunoreactivity.

  1. Lack of Analysis in Additional Brain Regions
    Limitation:
    The study is limited to the lateral nucleus of the amygdala, without consideration of potential receptor distribution differences across other limbic regions.

Recommendation:
Expand receptor mapping to additional brain areas involved in emotion and pain processing (e.g., basal and central amygdala, septum) to provide a more comprehensive neuroanatomical framework.

  1. Absence of Functional or Behavioral Correlation
    Limitation:
    The study is purely anatomical and descriptive, with no assessment of functional, pharmacological, or behavioral consequences of receptor activation.

Recommendation:
Integrate behavioral paradigms (e.g., anxiety or pain reactivity tests) before and after administration of cannabinoid receptor agonists or antagonists in future studies to establish structure–function relationships.

  1. Insufficient Methodological Detail for Immunoreactivity Analysis
    Limitation:
    The use of ImageJ for quantifying immunoreactivity is mentioned, but methodological details are lacking, and standardization of analyzed areas is not evident.

Recommendation:
Provide a clear description of image analysis criteria, including area selection, number of fields analyzed, threshold parameters, and reproducibility. Consider incorporating stereological methods for more accurate quantification.

  1. Absence of Negative Controls in Immunofluorescence
    Limitation:
    The study does not mention the inclusion of negative controls to validate the specificity of immunostaining.

Recommendation:
Incorporate appropriate controls, such as omission of primary antibodies (negative) and tissues with known expression of the target proteins (positive), to confirm the validity of immunohistochemical results.

  1. Unclear Functional Role of CB2R
    Limitation:
    The functional significance of CB2R in the central nervous system remains debated, yet the study refers to its expression as "moderate" without further investigation.

Recommendation:
Support morphological data with functional studies (e.g., treatment with β-caryophyllene, a CB2R agonist) or co-localization analyses with neuronal and glial markers to better characterize CB2R’s role.

  1. Lack of Direct Interspecies Comparison
    Limitation:
    Although the study references data from rodents, carnivores, and humans, it does not include a direct qualitative or quantitative comparison with these species.

Recommendation:
Include comparative tables or figures to highlight similarities and differences in receptor expression across species, thus enhancing the phylogenetic and translational significance of the findings.

Final Remarks

This study provides the first evidence of the presence of cannabinoid and cannabinoid-related receptors within the equine amygdala, laying the groundwork for the potential use of cannabinoid-based therapies in managing equine behavior and pain. However, to strengthen the scientific and clinical impact of these findings, it is crucial to address the outlined methodological limitations, integrate functional approaches, and broaden the anatomical and comparative context of future research.

Please add the following recent study on the use of cannabinol in horses to your bibliographic notes:

Interlandi C, Tabbì M, Di Pietro S, D'Angelo F, Costa GL, Arfuso F, Giudice E, Licata P, Macrì D, Crupi R, Gugliandolo E. Improved quality of life and pain relief in mature horses with osteoarthritis after oral transmucosal cannabidiol oil administration as part of an analgesic regimen. Front Vet Sci. 2024 Feb 6;11:1341396. doi: 10.3389/fvets.2024.1341396. PMID: 38379920; PMCID: PMC10876772.

Author Response

Reviewer #4

Critical Issues and Recommendations

  1. Limited Sample Size (n=6)
    Limitation:
    The small number of samples significantly reduces the statistical power of the study and limits the generalizability of the findings.

Recommendation:
Future studies should increase the sample size and include animals of varying age, sex, and physiological status to enhance the representativeness and robustness of the data.

Response: We thank the Reviewer for the suggestion. We acknowledge the limitation related to the small sample size (n=6) in the Discussion section (lines 353-360), which is primarily due to the difficulty in obtaining well-preserved equine brain specimens suitable for both histological and molecular analyses. We emphasized the need for future studies to include larger and more stratified cohorts based on age, sex, and physiological conditions. We agree that this would improve the robustness and translational value of the data, and we plan to pursue such directions in follow-up studies.

  1. Use of Formaldehyde-Fixed Tissue for RT-PCR Analysis
    Limitation:
    The use of paraformaldehyde-fixed samples likely compromised RNA integrity, resulting in the inability to detect TRPV1 and PPARγ transcripts.

Recommendation:
Conduct parallel analyses on fresh tissue or tissue preserved in RNA-stabilizing reagents (e.g., RNAlater®), or extract RNA from frozen samples to obtain more reliable gene expression data.

Response: We agree that the use of formaldehyde-fixed, paraffin-embedded (FFPE) tissue can severely compromise RNA integrity and yield, especially for targets with low expression levels such as TRPV1 and PPARγ. As stated in the Methods section, the fixed tissue used for RT-PCR in this study was the only material available, and our intent was to perform a preliminary screening to assess the detectability of transcripts in archival material. We fully recognize the limitations of this approach and have accordingly interpreted the negative RT-PCR results with caution. In future investigations, we plan to use fresh or RNAlater®-preserved samples to ensure higher RNA quality and enable a more reliable assessment of gene expression profiles.

  1. Discrepancy Between Gene Expression and Immunohistochemistry Data
    Limitation:
    The absence of TRPV1 and PPARγ mRNA despite evident protein immunoreactivity raises concerns about the reliability of the methods or sample preservation.

Recommendation:
Validate protein presence using complementary methods such as Western blotting on fresh tissue, and include rigorous negative and positive controls to confirm antibody specificity and immunoreactivity.

Response: We acknowledge the apparent discrepancy between the lack of detectable mRNA for TRPV1 and PPARγ and the presence of protein immunoreactivity. This mismatch is most likely due to the degradation of RNA in fixed samples, as discussed in our response to point 2, rather than a lack of gene expression. Conversely, protein antigens are generally more stable and detectable even in paraformaldehyde-fixed tissue, which may explain the preserved immunoreactivity.

Regarding antibody specificity, we did include standard negative controls in our immunohistochemical experiments (e.g., omission of primary antibodies), which showed no unspecific signal. Furthermore, we have taken several measures to ensure antibody specificity and reliability in our study, which we mentioned in the M&M section:

  • For the molecular part, specific primers for Cnr1, Cnr2, Trpv1 already used in equine tissue by Zamith et al. (2023), while primers sequence to detect transcript of PPARγ were reported in material and method section (paragraph 4.1). The specificity of the amplified PPARγ PCR products was confirmed by agarose gel electrophoresis and melting curve analysis (this sentence has been added in the text in material and method, paragraph 4.1).
  • For the immunofluorescence analysis, all antibodies were selected based on prior use and validation in the literature, especially on equine tissues. Specifically:
  • The anti-CB1R, -TRPV1, and -PPARγ antibodies had already been tested by our research group using Western blot analysis on horse tissues, as published in Chiocchetti et al. (2021) [80] and Galiazzo et al. (2021) [81].
  • The rabbit anti-CB2R antibody (PA1-744) had also been tested by Western blot on equine tissue in a separate study (Kupczyk et al., 2022) [82].
  1. Lack of Analysis in Additional Brain Regions
    Limitation:
    The study is limited to the lateral nucleus of the amygdala, without consideration of potential receptor distribution differences across other limbic regions.

Recommendation:
Expand receptor mapping to additional brain areas involved in emotion and pain processing (e.g., basal and central amygdala, septum) to provide a more comprehensive neuroanatomical framework.

Response: We thank the Reviewer for this valuable suggestion. In the present study, we focused on the lateral nucleus of the amygdala due to its established role in the integration of emotional and nociceptive inputs. We agree that a broader mapping of cannabinoid and related receptors across other subregions of the amygdaloid complex (e.g., other basolateral nuclei, central nuclei) and additional limbic structures (such as the septum or hypothalamus) would provide a more comprehensive understanding of their distribution and potential functions. We are currently designing follow-up experiments to explore these regions in the equine brain using fresh tissue and a combination of histological and molecular techniques. That said, we believe the Reviewer will appreciate that performing a detailed anatomical characterization of four receptor systems in a large animal species represents a time- and resource-intensive endeavor. For this reason, we decided to focus our investigation on one well-defined and functionally relevant area as an initial step.

  1. Absence of Functional or Behavioral Correlation
    Limitation:
    The study is purely anatomical and descriptive, with no assessment of functional, pharmacological, or behavioral consequences of receptor activation.

Recommendation:
Integrate behavioral paradigms (e.g., anxiety or pain reactivity tests) before and after administration of cannabinoid receptor agonists or antagonists in future studies to establish structure–function relationships.

Response: We appreciate the Reviewer’s insight. As correctly pointed out, this study was conceived as a neuroanatomical baseline investigation, aiming to provide the first evidence of the distribution of CB1, CB2, TRPV1, and PPARγ receptors in the equine amygdala. We fully agree that the functional and behavioural correlates of these receptor systems, particularly in relation to pain modulation and emotional regulation, represent an essential next step toward translational and clinical relevance. However, we would like to point out that it is ethically and practically challenging to obtain equine brains from animals previously treated with cannabinoid receptor agonists or antagonists outside of a therapeutic context. Such protocols would raise significant ethical concerns and are unlikely to receive approval from institutional animal care and use committees, especially in a species like the horse. Nevertheless, we are planning to investigate naturally deceased animals that were clinically monitored for pain- or behaviour- related disorders during life, in order to explore possible correlations between receptor distribution and pathophysiological conditions. We believe the reviewer will appreciate the complexity involved in collecting such samples, and acknowledge the current study as a necessary and foundational step for any future structure/function investigation.

  1. Insufficient Methodological Detail for Immunoreactivity Analysis
    Limitation:
    The use of ImageJ for quantifying immunoreactivity is mentioned, but methodological details are lacking, and standardization of analyzed areas is not evident.

Recommendation:
Provide a clear description of image analysis criteria, including area selection, number of fields analyzed, threshold parameters, and reproducibility. Consider incorporating stereological methods for more accurate quantification.

Response: We thank the Reviewer for highlighting this important aspect. We agree that a more detailed description of the immunoreactivity quantification procedure would enhance the clarity and reproducibility of our findings. In the revised manuscript, we have added specific methodological information regarding the number of fields analyzed per section, the anatomical localization of these fields within the lateral amygdaloid nucleus, the parameters applied in ImageJ (thresholding method, measurement settings), and the procedures used to ensure consistency across samples (see Sections 4.4 and 4.5). Regarding the Reviewer’s suggestion to consider stereological methods for more accurate quantification, we would like to clarify that in our study the immunoreactive (IR) signal was predominantly localized within the neuropil (i.e., axon terminals, dendrites), rather than in individual cell bodies. As such, classical stereological approaches, typically applied for unbiased cell counting, would be difficult to implement in this context. Instead, we adopted a standardized and consistent sampling strategy across the rostro-caudal axis of the lateral nucleus, analyzing IR signal density as a percentage of the total field area. This semiquantitative approach using threshold-based ImageJ analysis is widely accepted in the literature for describing the distribution and intensity of diffuse labeling patterns. All images were acquired under identical exposure settings and analyzed blindly by the same trained observer to ensure reproducibility. We hope that these clarifications and methodological additions will address the reviewer’s concerns.

  1. Absence of Negative Controls in Immunofluorescence
    Limitation:
    The study does not mention the inclusion of negative controls to validate the specificity of immunostaining.

Recommendation:
Incorporate appropriate controls, such as omission of primary antibodies (negative) and tissues with known expression of the target proteins (positive), to confirm the validity of immunohistochemical results.

Response: We thank the Reviewer for pointing out the importance of proper controls in immunofluorescence studies. As detailed in the Materials and Methods section (paragraph 4.3 “Specificity of the Antibodies”), we performed negative controls by omitting the primary antibodies and incubating the sections only with the secondary antibodies. Under these conditions, no immunoreactive signal was detected in any of the tested samples, confirming the absence of non-specific labeling. Moreover, all primary antibodies employed in this study have been previously validated for use in equine tissues by our group and others using Western blotting, as specified in response 3. We have now emphasized this point more clearly in the manuscript to prevent possible misunderstandings. We trust that this clarification demonstrates the reliability and specificity of the immunostaining procedures used in the present work.

  1. Unclear Functional Role of CB2R
    Limitation:
    The functional significance of CB2R in the central nervous system remains debated, yet the study refers to its expression as "moderate" without further investigation.

Recommendation:
Support morphological data with functional studies (e.g., treatment with β-caryophyllene, a CB2R agonist) or co-localization analyses with neuronal and glial markers to better characterize CB2R’s role.

Response: We appreciate the Reviewer’s comment regarding the ongoing debate on the functional significance of CB2 receptors in the central nervous system. Our study aimed primarily at providing a detailed neuroanatomical description of CB2R expression in the equine amygdala, rather than addressing functional implications. As such, we intentionally limited speculative interpretations and referred to the receptor distribution as "moderate" in order to remain within the descriptive scope of the study. We agree that further investigation is needed to clarify CB2R’s functional role in the horse brain. However, as also mentioned in our reply to point 5, pharmacological experiments involving administration of CB2R agonists or antagonists in horses for purely research purposes are currently limited by stringent ethical and regulatory considerations, which restrict experimental treatments in the absence of therapeutic indications. We agree that further investigation is needed to clarify CB2R’s functional role in the horse brain. To this end, we conducted preliminary co-localization experiments using GFAP alongside CB1R, CB2R, TRPV1, and PPARγ. However, we did not observe convincing glial co-localization signals for CB1R, CB2R, or TRPV1. Only PPARγ displayed evident immunoreactivity within GFAP-positive astrocytes. Based on these findings, we decided to include only the PPARγ/GFAP double-labeling results in the present study, and not to pursue further analysis with CB2R at this stage.

  1. Lack of Direct Interspecies Comparison
    Limitation:
    Although the study references data from rodents, carnivores, and humans, it does not include a direct qualitative or quantitative comparison with these species.

Recommendation:
Include comparative tables or figures to highlight similarities and differences in receptor expression across species, thus enhancing the phylogenetic and translational significance of the findings.

 Response: We appreciate the Reviewer’s suggestion to include a comparative table summarizing the available literature on cannabinoid and cannabinoid-related receptor distribution across species. While we understand the potential value of such a resource, after careful consideration and discussion with our co-authors, we believe that assembling a comparative table based on highly heterogeneous studies, often employing different morphological, molecular, and functional techniques, would risk being misleading rather than informative. In fact, the available literature varies widely in terms of anatomical resolution, detection methods (e.g., autoradiography, in situ hybridization, immunohistochemistry), quantification criteria, and even anatomical nomenclature, thereby limiting the possibility of a reliable one-to-one comparison. Nevertheless, in response to the reviewer’s interest in the interspecies perspective, we have added a dedicated section to the Discussion (lines 165–209) that thoroughly summarizes and contextualizes previous findings on CB1, CB2, TRPV1, and PPARγ receptors in the amygdaloid complex of other mammalian species, including rodents, humans, marmosets, and dogs. This narrative synthesis highlights key similarities and differences with the present results and emphasizes the novelty and translational relevance of our findings in the horse.

Final Remarks

This study provides the first evidence of the presence of cannabinoid and cannabinoid-related receptors within the equine amygdala, laying the groundwork for the potential use of cannabinoid-based therapies in managing equine behavior and pain. However, to strengthen the scientific and clinical impact of these findings, it is crucial to address the outlined methodological limitations, integrate functional approaches, and broaden the anatomical and comparative context of future research.

Response: We thank the Reviewer for the constructive and detailed feedback. We agree that this study, while primarily anatomical and exploratory, provides an essential foundation for future investigations on the role of cannabinoid and cannabinoid-related receptors in the equine brain. As the first report documenting the expression of CB1, CB2, TRPV1, and PPARγ receptors within the equine amygdala, our findings lay the groundwork for translational research aimed at understanding the neurochemical basis of behavior and pain modulation in this species. We acknowledge the limitations related to sample size, methodological constraints, and the absence of functional or behavioral data. However, we hope the reviewer will appreciate the novelty, technical complexity, and ethical considerations involved in this type of neuroanatomical study in large animals. We believe that addressing all the outlined aspects in a single study would have gone beyond the scope and feasibility of the present work. Nonetheless, we have taken care to discuss these limitations transparently and to outline future directions that may enhance the scientific and clinical relevance of our findings.

Please add the following recent study on the use of cannabinol in horses to your bibliographic notes:

Interlandi C, Tabbì M, Di Pietro S, D'Angelo F, Costa GL, Arfuso F, Giudice E, Licata P, Macrì D, Crupi R, Gugliandolo E. Improved quality of life and pain relief in mature horses with osteoarthritis after oral transmucosal cannabidiol oil administration as part of an analgesic regimen. Front Vet Sci. 2024 Feb 6;11:1341396. doi: 10.3389/fvets.2024.1341396. PMID: 38379920; PMCID: PMC10876772.

Response: We thank the Reviewer for this valuable suggestion. As requested, we have added the reference to the study by Interlandi et al. (2024), which reports improved quality of life and pain relief in mature horses with osteoarthritis following oral transmucosal administration of cannabidiol. To further contextualize the potential translational relevance of our anatomical findings, we have also integrated additional recent clinical studies exploring the use of cannabinoids in horses affected by pain, inflammation, and behavioural disturbances (Zamith Cunha et al., 2023; Turner et al., 2023; Ellis and Contino, 2019; Aragona et al., 2024). These references are now included in the final part of the Conclusion section (lines 491-494), to better support the rationale for future functional investigations.

Round 2

Reviewer 2 Report

Comments and Suggestions for Authors

The author has made the revisions according to my suggestion, so I agree to accept it in its current form.

Reviewer 4 Report

Comments and Suggestions for Authors Dear authors, the manuscript is much improved and can be accepted in the present form.